# Lessons Learned from Topic Modeling Analysis of COVID-19 News to Enrich Statistics Education in Korea

**Seokmin Kang [1] and Sungyeun Kim [2],***

[1] College of Education and P-16 Integration, the University of Texas Rio Grande Valley, 1201 W University Dr, Edinburg, TX 78539, USA; seokmin.kang@utrgv.edu

[2] Graduate School of Education, Incheon National University, 309Ho 15Hokwan Academiro 119, Incheon 22012, Korea

* Correspondence: syk@inu.ac.kr; Tel.: +82-32-835-8163

**Abstract:** This study aimed to investigate how mass media in Korea dealt with various issues arising from COVID-19 and the implications of this on statistics education in South Korea during the recent pandemic. We extracted news articles with the keywords "Corona" and "Statistics" from 18 February to 20 May 2020. We employed word frequency analysis, topic modeling, semantic network analysis, hierarchical clustering, and simple linear regression analysis. The main results of this study are as follows. First, the topic modeling analysis revealed four topics, namely "macroeconomy", "domestic outbreak", "international outbreak", and "real estate and stocks". Second, a simple linear regression analysis displayed two rising topics, "macroeconomy" and "real estate and stocks" and two falling topics, "domestic outbreak" and "international outbreak" regarding the statistics related to COVID-19 as time passed. Based on these findings, we suggest that the high school mathematics curriculum of Korea should be revised to use real-life context to enable integrated education, social justice for statistics education, and simple linear regression analysis.

**Keywords:** COVID-19; educational sustainability; text mining; topic modeling; statistics education; Korea

## 1. Introduction

Since the outbreak of COVID-19 was first reported in China, the agendas for political, economic, social, and cultural issues have been interpreted in relation to the COVID-19 pandemic. Initially reported in Wuhan as pneumonia caused by an unknown pathogen, COVID-19 gradually spread to other provinces of China and all over the world. The World Health Organization declared the pandemic as the highest level of emergency. This contagious disease has intimidated humankind worldwide, thereby raising concerns about potential chaos and economic losses. Because of the pandemic, education is also under a state of emergency [1,2].

This unprecedent unpredictability calls for appropriate responses and follow-up measures to manage this crisis. To respond to the request, it is necessary to systematically integrate accumulated data and to have the ability to interpret findings based on data analysis. Mathematics lies at the center of fundamental societal reform and should be able to solve complex problems that are closely related to sustainability—such as manufacturing and security—as well as social problems—such as disaster management, modeling of infectious diseases, crime prevention, transportation, environmental conservation, and welfare—through objective and accurate predictions based on big data. In particular, to prepare for another pandemic in the future, it is necessary to have big data and accurate information. In this sense, teaching statistics in mathematics curriculum functions as the basis for analyzing and predicting outbreaks and is becoming increasingly essential [3].

Statistics has received attention and is emphasized in formal education, as it includes cognitive tasks such as problem solving, data analysis, and data interpretation [4,5]. For

example, the 2015 revised mathematics curriculum in South Korea emphasizes that statistical literacy is required in the process of collecting, summarizing, and interpreting data as basic literacy for every democratic citizen in order to predict the future and make rational decisions by understanding the uncertainty of the modern information society [6]. However, unlike other advanced countries in North America and Europe, as well as Japan and China, which have built a strategy to prepare for an unpredictable future and strengthen mathematics education by revising their school curricula [7–9], it is somewhat concerning that an educational policy in its curriculum in South Korea showed reduction in content related to mathematics (to lessen students' academic burden). For instance, students in the U.S. who enter colleges learn regression, Chi-square analysis, one- and two-tailed tests, t-distribution, parameter estimation and margin of error, unbiasedness, and point estimate among others through their AP courses. In addition, the students in China, England, Singapore, and Australia deal with regression analysis [7,9].

Another concern regarding statistics education in South Korea is that relatively more statistical skills are taught mechanically without considering their practical use [10]. Statistical education in South Korea only focuses on teaching data sorting skills, statistical calculation, and probability distribution theory through artificial examples without intellectual challenges, instead of "real world statistics", which is a tool to understand and predict real-world events [11]. This trend would be a threat to national competitiveness. Accordingly, the Second Comprehensive Mathematics Education Plan prepared students for process-oriented instructional methods focusing on principles and conceptions rather than problem solving based on rote memory, and it sought a paradigmatic shift in mathematics education by focusing on mathematical usefulness through examples from real life [12]. It has also been argued that statistics in the Fourth Industrial Revolution era should be reborn as an integrated subject with big data analysis, while teaching students the overall process of data collection, visualization, interpretation, and prediction, instigating them to have an insight in data from various perspectives [9,13].

One of the ways to ensure effective mathematics education is by using the statistics project method, in which the students themselves focus on selecting subjects, collecting and analyzing data, and interpreting the results. This should be administered as an integrated learning activity with various methods and plans under one topic. Through this, the students shall be able to experience the whole process of statistics [14–16]. Furthermore, the 2015 revised mathematics curriculum in South Korea defined information processing competency as collecting, organizing, analyzing, interpreting, and utilizing various data and information and processing information effectively by selecting an appropriate teaching tool. In mathematics curriculum, you can use the statistics project learning method as a way of strengthening information processing competency. Specifically, Wild and Pfannkuch [17] suggested five stages—problem, plan, data, analysis, and conclusion—for how people think and what kind of behavior they perform throughout a statistical survey. Since then, Franklin et al. [14] adjusted it into four stages: problem setting, data collection, data analysis, and result interpretation. Although implementing statistics curriculum in the school system alone is insufficient to guarantee mathematics education reform, the statistics project method can have a great impact on helping students transform their inertial thinking styles, such as relying on rote memorization and solving problems mechanically, into active and critical thinking.

In this study, we use a topic modeling analysis as a testing bed that manifests how the statistics project method is designed and implemented to train our students' problem-solving skills and broaden their perspective to prepare for an unpredictable future. Topic modeling is a kind of a probabilistic generative modeling used widely in many fields with a specific focus on text mining and information retrieval [18–20]. Recently, many studies using topic modeling based on big data are actively administered. Among these are a scientific issue grasping a trend of new renewable energy [21], an economic issue dealing with an economic policy uncertainty [22], an entertainment issue about popular dramas [23], an education issue dealing with gifted education [24], a political or foreign affair related issue

such as Terminal High Altitude Area Defense (THAAD) deployment [25,26], a cultural issue dealing with the study of international happiness and unhappiness [27], a natural disaster issue such as earthquake or fine dust [28,29], a life issue about conflict between residents in France and wild beasts [30], a policy issue related to currency and monthly price developments reports by banks [31], a social issue about the Fourth Industrial Revolution [32], an ecological issue of zoonotic diseases transmitted from non-human mammals [33], and so on. These studies used topic modeling to analyze various issues holistically with automated big data analytics applicable to large-scale texts. Therefore, the focus of the present study rests on educational sustainability in Korea. Various thinking skills embedded in statistics project methods are closely related to preparing for an unpredictable future. Here, we show topic modeling as a testing bed to demonstrate how statistics project methods can be used by adopting a topic during the COVID-19 era.

Specifically, the study intends to determine implications for South Korean statistics education based on a statistics project class by performing topic modeling on data including "Corona" which is an official term in Korean language referring to COVID-19 and "Statistics" as atypical text material.

It focuses on the following research questions:

1.  How were COVID-19 statistics presented in the mainstream media?

    - What kinds of statistics were frequently used in the context of the COVID-19 outbreak?
    - What were the topics of interest in the news articles containing "Corona" and "Statistics"?
    - How did the trend of topics change over time?

2.  What implications for South Korean statistics education can be drawn from statistics used in the social world during the COVID-19 diffusion?

## 2. Literature Review

### 2.1. Text Mining: Focusing on Topic Modeling

Text mining is one of the methods of analyzing unstructured data. It is a process of analyzing a document collection with a variety of analysis tools [34]. The rise of computing power and advancement in algorithm has led to the development of information retrieval and natural language processing. In social science, content analysis has been one of the main techniques of understanding text data and applying it to many different forms of documents; it includes the analysis of propaganda in communication, myths and oral folktales in anthropology, and historical archives in history [35]. In the 2000s, text data analysis developed to a new phase of computerization. Combined with corpus linguistics, storing a large volume of data had far fledged in the 1990s. When machine learning became a leading technology to dissect and analyze natural language around the 2000s, categorizing and clustering documents and extracting information through the computerization method became one of the standard methods to understand text data and started to be applied in the social sciences.

Topic modeling is used to analyze text data with probabilistic understanding of documents. Given that a computer is unable to understand texts as humans do, computerization of text data requires transforming text data into mathematical calculations. This transformation employs statistical modeling for finding useful patterns of text data rather than a grammatical rule-based analysis. For example, latent semantic analysis uses an array of vectors to contextualize the relationship between words and infer the meaning of text [36]. The basic idea is to locate words in the vector space of words and calculate their probabilities of co-occurrence with the concept of similarity. Topic modeling uses the vector space model to understand texts. It is based on a bag-of-words algorithm to search, cluster, classify, and allocate documents into the collection of words called "topic" [20,37].

According to DiMaggio et al. [38], topic modeling has three advantages over the traditional text analysis methods such as content analysis. First, topic modeling can be replicated. Content analysis requires professional interpretation by experts to understand text, which would not be repeated by others differing in knowledge and intuition. Second,

topic modeling can handle massive amounts of text data with automation. Because topic modeling uses computerized algorithm, it does not require humans to understand the complex contexts of each document. Third, topic modeling allows researchers to take a break from theorizing or hypothesizing the content of texts before their analysis because it uses computer programs and assumes that understanding texts can be possible through the relation between words.

In topic modeling, a document is composed of the topic, "a distribution over a fixed vocabulary" [18]. For example, Latent Dirichlet Allocation (LDA) has been widely adopted for topic modeling methods in recent years [18–20]. LDA assumes there are multiple topics in a document. Each word in the document is thought that it is originated from a specific topic. Therefore, the modeling is based on the estimation of the probability that a certain word belongs to a specific topic. The computer algorithm finds the optimal probability to allocate words to a specific topic [39].

### 2.2. Topic Modeling Analysis Related to Mathematics Education

Using topic modeling, few studies have explored and shown a major trend in mathematical education. Notably, Inglis and Foster [40] traced the development of mathematical education research and collected research papers from the past 50 years (1968–2015) from two major mathematics education journals, comprising more than 3935 articles. A topic modeling tool to calculate possible topic models narrowed topic names down to 28, each of which corresponded to a different research article that contained the highest proportion of the topic name. Under each topic name were the top 20 characteristic words, which showed the kind of word associated with a certain topic name. For example, papers that were indicated to have "experimental designs" as a high proportion word had characteristic words such as test, study, group, scores, research, items, table, mathematics, tests, significant, results, groups, variables, analysis, treatment, ability, performance, item, journal, and experimental. Instead of reading a whole research article, one can quickly grasp what the research is about and what kind of ideas are mostly discussed mainly by focusing on the keywords. Throughout the topic model, Inglis and Foster [40] showed how the prominence of each topic had changed over time and changes in average topic proportions per year, and changes in research programs over 50 years.

Choi and Kwak [41] applied LDA to obtain the topics of mathematics education. They collected a total of 2556 article abstracts from 1990 to 2018 from *The Journal of Mathematical Behavior*, *The Journal for Research in Mathematics Education*, and *Educational Studies in Mathematics*; the number of topics was selected using deviance information criterion (DIC), where five topic models came out. It was noted that they summarized the topic structure of the journals every five years, which enabled us to keep track of the research trend systematically throughout the years. Jin and Ko [42] administered topic modeling on 470 research articles from 2016 to 2018 and investigated research studies in mathematics education in South Korea to observe a recent research trend. By combining frequent words together using Latent Dirichlet Algorithm (LDA) aided by intertopic distance map (IDM), they could narrow them down to five topic names which directed us toward what kinds of research topics have been dealt with. They found that the words with high frequency throughout the years from 2016 to 2018 were preparation, middle school, students, theory, acknowledgment, general, geometry, chance, and characteristic from topic 2, and evaluation, primary school, function, item, problem solving, relation, modeling, tool, calculation, and usage from topic 4. By comparing the previous studies using a trend analysis, they also found that "educational curriculum" was the main research topic since 2010, which had a different pattern from the previous one. Even though it is not the focus of this study, it would be interesting to compare the trend of research through topic names from different education systems from the review of the previous studies and how the results of the research trend are different between topic modeling based on abstracts and entire articles.

The literature review demonstrates that topic modeling in mathematics education is mainly focused on research articles to explore the current state and trends of research.

However, topic modeling of news articles can also provide practical knowledge, which represents what people have on mind and shows the trend of public interest or concerns, which in turn can be used as a strong tool for students to explore an idea, collect data, analyze the data, and interpret the results. Hence, in this study, we performed topic modeling analysis on news articles containing "Corona" and "Statistics" to provide implications for statistics education using statistics project class as an example led by students' activities through data collection, analysis, and interpretation.

## 3. Data and Methods

### 3.1. Data

We used the *BigKinds* news database service provided and maintained by the Korea Press Foundation (KFP). KFP is a semi-governmental organization under the Ministry of Culture, Sports and Tourism (MCST). MCST supervises the operation fund of KFP. *BigKinds* archives news articles from 54 news daily outlets, including five broadcasting channels and two ICT-specialized newspapers. The dataset provided by *BigKinds* contains the name of the news outlet, the reporting date, the section of the article, the entire content of news article as raw data and the refined word level trimmed document for the ease of text mining analysis.

We extracted news articles using a combination of the two search keywords, "Corona" and "Statistics" from 18 February to 20 May 2020 and treated each article as a document. This procedure produced 3815 documents. The first confirmed case of COVID-19 was identified in Daegu, Korea, on February 18. February 18 was the first day that the confirmed case of COVID-19 was identified in Daegu, Korea, where the massive virus contagion was initiated, leading to the three-month-long first massive infection wave. Korea experienced dramatic changes during the next 94 days. The first three months of the surge of infections were the most critical in the sense that the nation experienced a global pandemic. We believe that it is sufficient to collect data for these three months to understand how COVID-19 impacted the country. In early March, Korea was reported as the second largest country with laboratory confirmed COVID-19 cases around the world and then by the middle of March it became a model country that successfully controlled the virus diffusion without compulsory social distancing orders from the government [43]. We used this time window for the text analysis related to COVID-19 and statistics, as we consider it to be the most critical and suitable period for data collection.

### 3.2. Methods

We employed two data analysis methods. First, the text mining method [20,37] was used to understand the keywords and issues reported in the news articles. We have particularly used a standard procedure of text mining consisting of word frequency analysis, topic modeling, and semantic network analysis. Text mining enables you to identify patterns and relationships which exist within a large body of information [44]. The analysis of word frequency used two key indexes called term frequency (TF) and inversed document frequency weight on TF (TF-IDF). Whereas TF is a simple account of word frequency in the dataset document, TF-IDF uses document frequency of word appearance as a weight to calculate the importance of words. This index is made to prevent bias of treating simple word frequency as a single measure. For example, it is possible that the word frequency of "virus" could be at the highest level while it only appears in a few documents. If we simply consider TF in this case, the word "virus" in all the documents would take too much importance compared with its appearance proportion at the document level. Another example is related to the common word in the document. If a word appears in every document of analysis dataset, the word is of no use to distinguish a certain subject. In our case, the word "Corona" is this type of word as our searching words include the word "Corona" in the first place. In other words, the TF-IDF index is a measure to determine the interest level of the word in the document [45].

As for topic modeling, we used Latent Dirichlet Allocation. LDA has two assumptions. First, it assumes that the document has multiple topics and latent topics as the distribution of words are randomly dispersed in the document [18,46]. In other words, it basically calculates the probability that a bag-of-words belongs to a topic by using Dirichlet distribution parameters [18]. LDA topic modeling requires that the number of topics be set by researchers in advance.

On the aspect of network analysis, we applied semantic network analysis [47,48]. The network is constructed by the following procedures. First, we selected words that cover 95% of word frequency. Following this criterion leaves us with 399 words. Second, we made a documents-by-words matrix, a two-mode network, and transformed it into a word-by-word co-occurrence matrix using correlation of the co-occurrence between words. Third, we dropped any network tie less than the level of 0.1 correlation. Considering the sparsity of the matrix cells, this criterion is a reasonable assumption to delineate word relations. After following this process, the final matrix has 80 keywords and we analyzed the network from this matrix.

We strategically selected two techniques involved in network analysis to improve the accuracy of the results [49–51]. First, we used one of the network community detection algorithms in order to identify word clusters where the data have the most significant similarity [49]. The network community detection is different from topic modeling in the sense that it indicates the direct co-occurrence between words so that the network community is in fact densely connected; this means structure other than a bag containing words having different probabilities of belonging to a topic. We used the fast greedy algorithm to detect network communities introduced and formulated by Newman [52]. Second, we used cluster dendrogram analysis constructed by structural equivalence indexes. Structural equivalence calculates the similarity of network relations between ties. Differently put, if two nodes in a network have the identical network connections to other nodes in the whole network, there is no difference in their network position and role. Thus, they have the same equivalence score. If any relation is different, it leads to difference in their position and role in the network. The structural equivalence dendrogram draws the similarity between nodes based on their connections, hierarchically clustering the similarity. By clustering words based on this similarity, we can interpret how word clusters are constructed in sequence through the hierarchical clustering.

The other analysis method used in this study is simple linear regression analysis. To analyze whether each topic is increasing or decreasing in popularity, we performed a simple linear regression model for each topic with the date of news articles being independent variables and the topic proportions in the corresponding dates being dependent variables [53]. This regression analysis estimates whether a topic proportion increases or decreases as time passes. The unit of the regressor variable is a single day. In addition, we used the R software to conduct the analysis and the R software and UCINET to visualize our results.

## 4. Results

### 4.1. Keyword Analysis from Term Frequency

Our first keyword frequency analysis shows that the combination of "Corona" and "Statistics" relates to the report on the number of infected people around the world, including Korea, and the economic downturn, particularly related to industry sections and small and medium size Enterprises (SMEs). Table 1 displays TF and its inversed document frequency TF-IDF. In the first column of the TF section, the most frequent words are related to the outbreak statistics. They are words such as "confirmed case", "death toll", "mask", "jobs", "Italy", "global", "Trump", and "workers". In the next column of the TF section, "real estate" and "apartment" take relatively higher ranks in terms of frequency, but the main theme of word cluster in the column is "economics" with words such as "manufacture", "service", "car", "GDP", and "self-employed", This can be interpreted as

macroeconomic indexes. In the next column, the words "consumer", "SMEs", "employer", and "tourists" indicate statistics related to economics in smaller sized businesses.

**Table 1.** Word Frequency: TF and TF-IDF (Rank Order).

| TF | | | | | | TF-IDF | | | | | |
|---|---|---|---|---|---|---|---|---|---|---|---|
| corona | 18,827 | stakeholders | 1054 | contagious disease | 574 | confirmed case | 9828.8 | story | 3155.8 | infection | 2244.4 |
| confirmed case | 8373 | online | 944 | local time | 566 | death toll | 7118.1 | manufacturing | 3011.0 | midsize companies | 2242.8 |
| death toll | 3905 | real estate | 941 | Midsize companies | 556 | mask | 6310.9 | Russia | 2916.5 | MERS | 2160.8 |
| last month | 2677 | Statistics Bureau | 920 | Chinese | 540 | President | 6197.4 | foreigner | 2835.1 | infectious disease | 2094.1 |
| virus | 2564 | manufacturing | 887 | Jobless | 536 | jobs | 5638.2 | Statistics Bureau | 2831.2 | unemployed | 2092.9 |
| president | 2455 | service | 864 | International | 534 | Shincheonji | 5105.4 | infected disease | 2745.4 | democratic party | 2068.7 |
| mask | 2200 | infected disease | 834 | hardship | 534 | apartment | 4671.0 | data | 2702.3 | next | 2027.9 |
| jobs | 2003 | story | 814 | pandemics | 531 | Trump | 4665.2 | service | 2672.6 | laborer | 1986.6 |
| infection | 1968 | France | 804 | MERS | 520 | workers | 4660.5 | France | 2667.5 | employee | 1943.5 |
| people | 1649 | expert | 768 | committee | 517 | Italy | 4563.4 | Spain | 2638.3 | pandemics | 1942.2 |
| possibility | 1642 | foreigners | 759 | laborers | 491 | virus | 4546.7 | stakeholders | 2530.1 | President Moon. | 1933.9 |
| Italy | 1565 | Spain | 738 | increase | 491 | people | 4085.8 | Chinese | 2502.4 | SMEs | 1918.5 |
| infected person | 1534 | data | 701 | companies | 471 | last month | 4080.8 | service industry | 2436.1 | consumer | 1913.4 |
| global | 1360 | service industry | 660 | citizens | 469 | infected person | 4076.8 | healthcare workers | 2365.2 | Arrivals | 1907.6 |
| Trump | 1318 | minus | 656 | consumers | 450 | real estate | 3684.9 | corona | 2349.9 | citizens | 1886.1 |
| Shincheonji | 1256 | car | 625 | SMEs | 435 | possibility | 3463.6 | GDP | 2321.5 | bonds | 1837.6 |
| Workers | 1256 | Russia | 623 | direct hit | 434 | global | 3404.1 | minus | 2310.6 | hardship | 1825.4 |
| apartment | 1172 | Healthcare workers | 617 | briefing | 431 | national | 3343.3 | self-employed | 2273.1 | patients | 1785.2 |
| National | 1092 | GDP | 614 | employee | 431 | online | 3223.2 | expert | 2261.4 | international | 1765.7 |
| points | 1068 | self-employed | 588 | democratic party | 428 | point | 3207.6 | car | 2254.1 | IMF | 1762.9 |

Note: TF: term frequency; TF-IDF: TF-inverse document frequency; GDP: gross domestic product; MERS: Middle East respiratory syndrome; SME: small and medium-sized enterprises.

In the TF-IDF section of Table 1, which weighs the document frequency in the decision of term-frequency rank, a similar pattern is found, but the ranks of "apartment" and "real estate" jumped up compared with the TF section. The impact of the virus tends to also be linked with the "real estate" housing price in some ways, and more importantly, it is more widely shared by newspapers, more than the global trend of pandemic diffusion. The ranked places of other words seem to be similar to the TF section. The top rank word "Corona" in the TF section has disappeared in the TF-IDF section, indicating it is the most common word in every document.

Some interesting points are worth mentioning. First, the words "workers" and "jobs" are more frequent than the word "unemployed". The rank of the term "unemployed" is far lower than that of "workers" and "jobs". From the viewpoint of statistics, they are correlated, but the difference in the nuance can be postulated. It seems that the emphasis of statistics is given to creating or at least maintaining the number of workforce and "jobs" rather than the protection of "unemployed". Second is the relation between the "real estate" market, particularly, "apartment" the most prevalent form of housing in Korea, and it is possible to interpret the results as people being concerned most about their assets that could be influenced by the pandemic, given that the weight of household assets in Korea is significantly high by real estate when compared with the ones in the U.S., England, Australia, and The Netherlands [54]. While it is reasonable to consider the impact of macro indexes of economics, the impact of the pandemic on the real estate market needs to be further scrutinized. Third, it is rare to see statistics on the health care system; all statistics are related to economics. The negative impacts of COVID-19 on societal response systems such as the shortage in the number of beds or ventilators was not the issue for South Koreans. Overall, the frequent word usages during the outbreak of COVID-19 reveal that the most interested topic theme lies in economics and the number of laboratories that confirmed cases.

While this manifests the obvious connection between the economy and the pandemic, more elaboration is required on how these connections are described and constructed as a narrative. In the subsequent analysis, we trace the actual content of each topic and examine the narrative on how statistics are utilized to make a story.

### 4.2. LDA Topic Model and Semantic Network Analysis

The LDA topic model and semantic network analysis describes the detailed narrative related to statistics and COVID-19. Figure 1 depicts the probability of the top 20 words belonging to each topic.

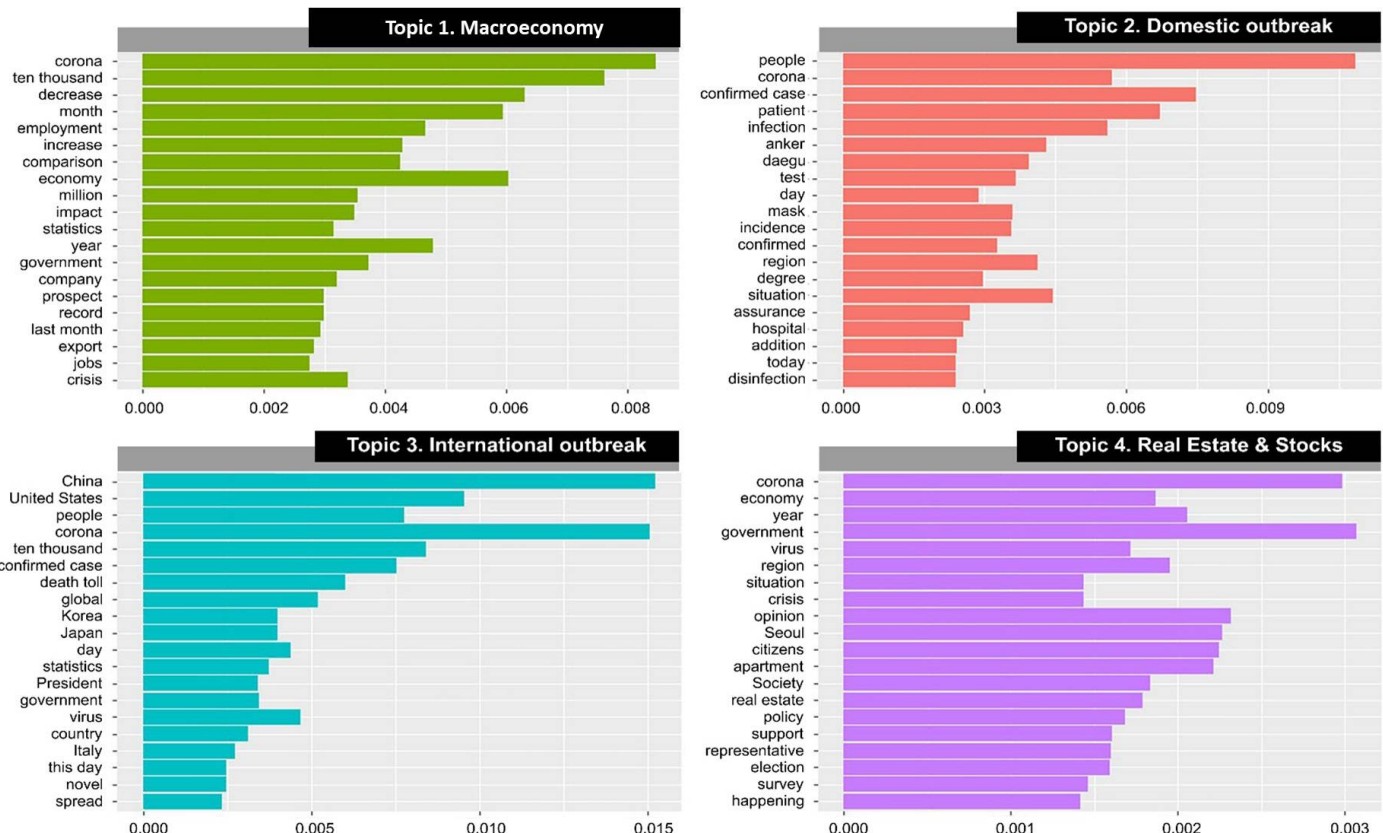

**Figure 1.** LDA topic model result: top 20 word probabilities to topics.

The most interesting feature in Figure 1 is the macroeconomic topic. It contains the past trends, current status, and future prospects at the same time. This triangular description of the past, the present, and the future indicates the typical storyline of economic statistics, particularly for a country such as Korea in which "export" is the main driver of economic growth. Furthermore, during the pitfall of the stock market due to COVID-19, individual holdings have significantly increased because of the increase in the topic. While the experts argued that the risk of holding stocks in the stock market increased under COVID-19, individual holdings have been significantly augmented. For example, the number of new individuals holding Samsung Electronics stock has increased to more than one million, a 154% increase during three months from February to April 2020 [3]. The interest in the economy may be related to "crisis", but it is more than an interest in the business cycle.

The outbreak statistics can be divided into domestic (Topic 2) and international (Topic 3) outbreak interest. However, there is a clear difference between the two at the unit level. In the domestic topic, the simple counting unit "number of people" has the highest probability belonging to the topic, whereas it is "ten thousand number of people" for the internationally confirmed cases. Besides, the death toll has a place in the international statistics, but it is more about "incidences" rather than "deaths" in Korea. Indeed, Italy has

one of the highest mortality rates due to COVID-19. It was 14.3 percent in Italy, whereas it was 2.3% in Korea as of 29 May 2020 [55], thus the severity in mortality rate is reflected in the statistics.

The last topic, "real estate and stocks", reveals that this aspect was regarded as a general election issue because the data included the period of an election for the National Assembly. Statistics related to housing price and government policies are consolidated as the main issue of the election during the spread of COVID-19. Therefore, these statistics are intriguing, as the election issue has come down to how government has responded to the COVID-19 outbreak and controlled the market price of real estate. COVID-19 is therefore one of the current affairs discussed in the newspapers.

Figure 2, the semantic network diagram, precisely represents this interest in real estate. In Figure 2, the words "real estate" and "apartment" are far aside from the main cluster of the network community that the two words belong to. Semantic network indicates the direct co-occurrence between words. Compared with the topic model—a basket model that put related words in one basket—the network approach draws a direct connection between words for analysis interpretation. Thus, the relative distance from the main core cluster of the community means that it comprises a separated domain even though it is a part of the network community.

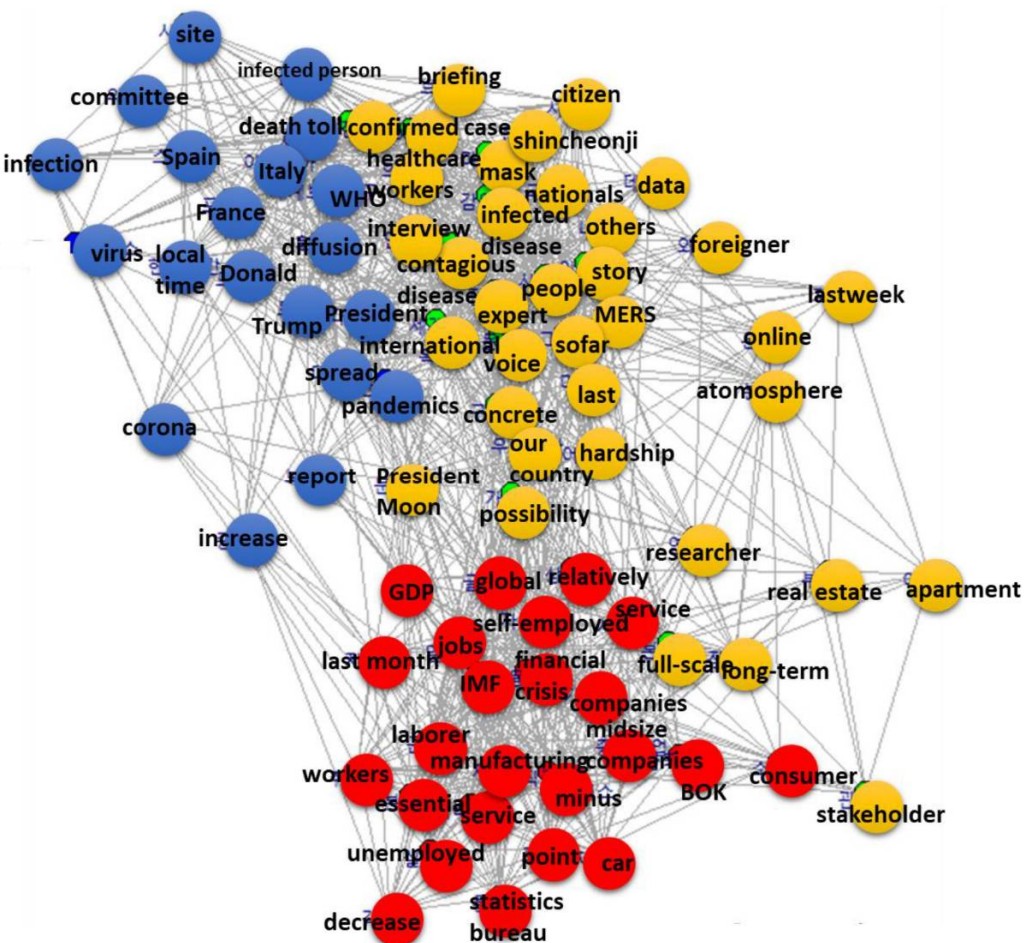

**Figure 2.** Semantic Network.

It is interesting that real estate is the main economic issue for the general election. People invest lots of money in the stock market during the pitfall of a crisis such as the current pandemic. In addition, housing is a primary issue for the election. Statistical education to analyze macroeconomic indexes and housing market should then have become

an essential deliverable, but statistical education largely remains in the theoretical domain rather than the practical.

Other network communities exhibit a pattern similar to that explained in the LDA topic model, except the macroeconomy topic. Unlike the LDA topic model, the network communities show the connection between all agents involved in the economic activities such as "Bank of Korea (BOK)", "small size companies", "global", "workers", "jobs", "GDP", "unemployed", and many others. However, *chaebol*, a large conglomerate of family-owned companies is still missing in the network. It seems the big players in the Korean economy are robust, except the "car" industry, which is directly impacted by social distancing.

### 4.3. Clustering with Structural Equivalence Measure

We analyzed the data one step further to delineate how stories in mainstream media are constructed. The hierarchical dendrogram (Figure 3) indicates six different stories related to the COVID-19 pandemic and the statistics. They are domestic economics, global economy and Korea, real estate and consumption, stock market, global diffusion of the virus, and domestic policy of disease control.

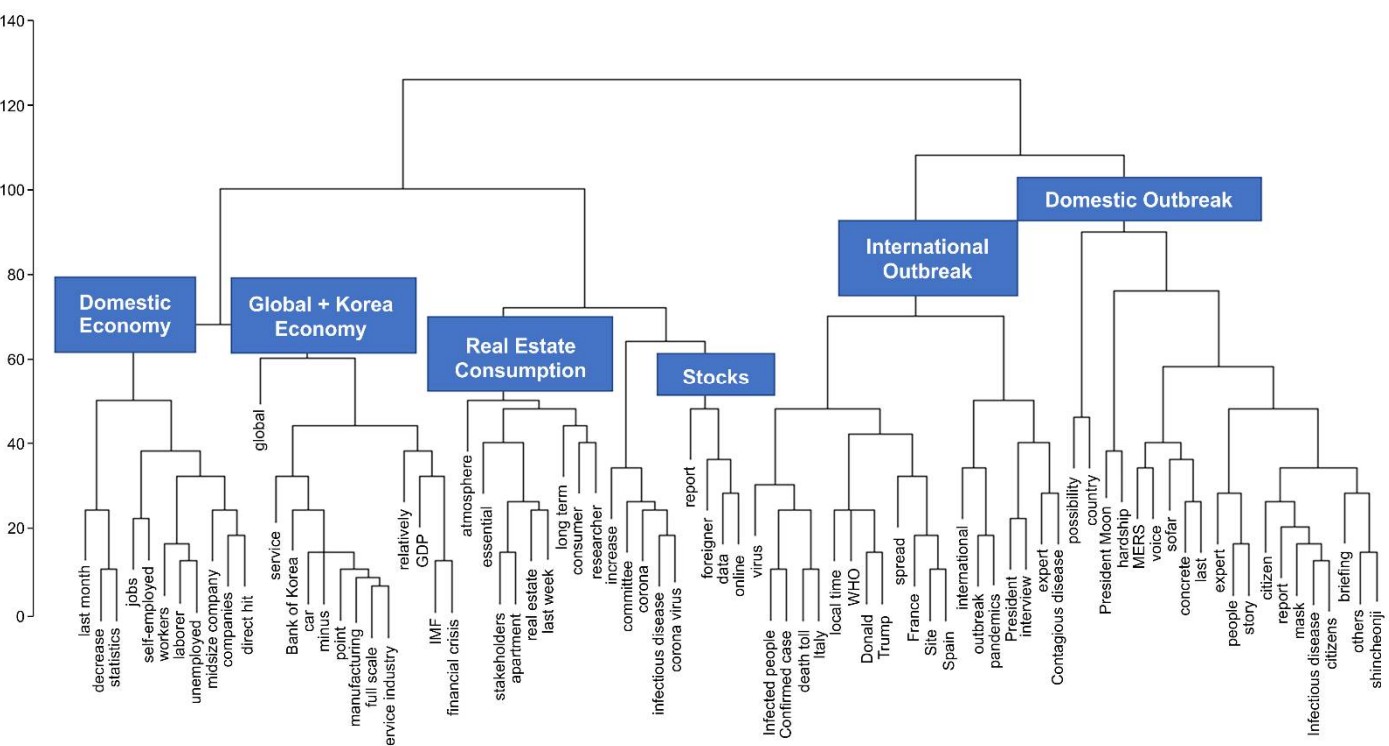

**Figure 3.** Structural Equivalence Cluster Dendrogram.

Figure 3 contains three main points. First, now it is revealed that the large chaebol business is discussed at the global level, as we can see the word manufacturing sector is connected to service sector and then it meets the car industry and BOK. The words "financial crisis" and "IMF" meet the word "GDP" in the diagram tree, then it becomes a part of the global economy and the Korea section. Second, the word "data" is related to the stock market report, particularly the move of foreign investors, one of the most influential agents in the stock market. Third, two different strands in the global diffusion of virus shows that "expert" and "interview" and their prediction are related to our search word—"statistics".

*4.4. Trend in Topics: Simple Linear Regression Analysis*

For the last analysis, we performed a simple linear regression analysis. We used the date of news articles as the independent variable and conducted the influence of date in topic terms. According to our analysis, the statistics related to COVID-19 show that the confirmed cases statistics, both domestic and international, tend to decrease over time, while the economics topics take an upward trend during the same period, as shown in Figure 4. In addition, they are statistically significant, while differing in significant levels and slope of regression coefficients. To illustrate, "real estate and stock", which is a topic related to the general election, tends to increase rapidly. Statistics of domestic confirmed cases are quite the opposite, reflecting the swift flattening of the diffusion curve in Korea.

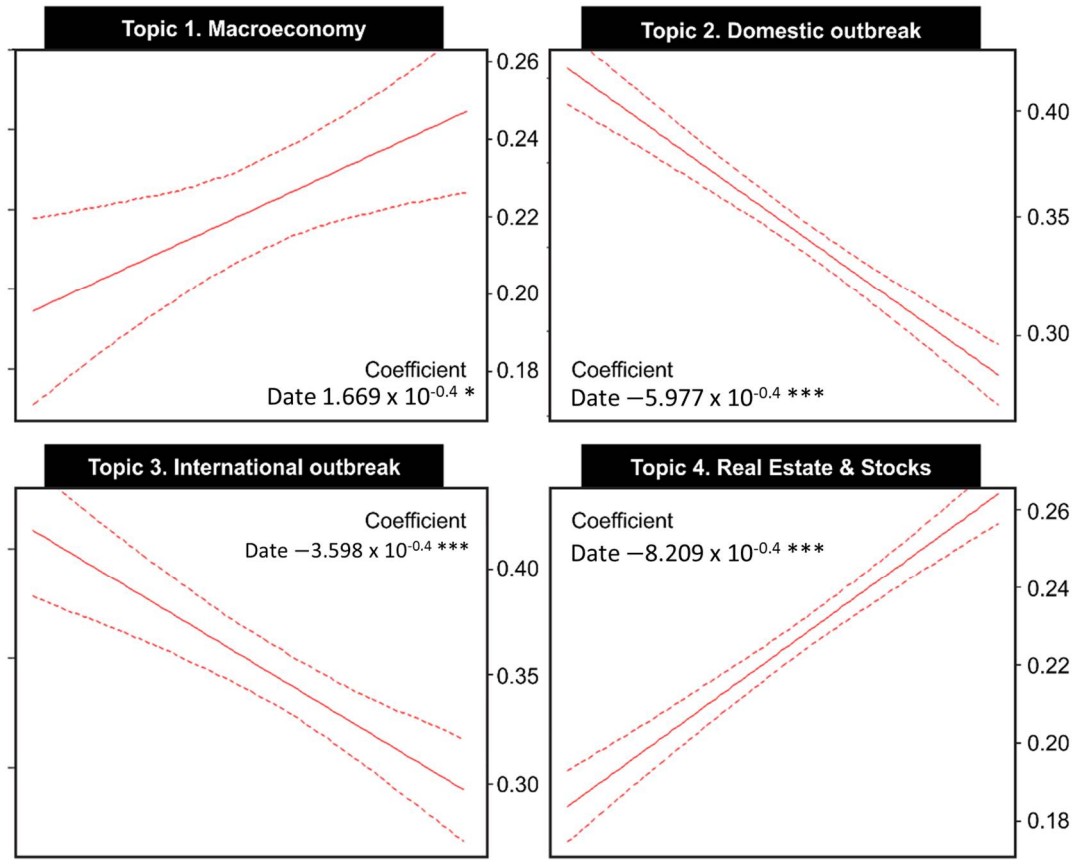

**Figure 4.** Results of simple linear regression analysis. * $p < 0.05$, *** $p < 0.001$. X-axis: the date from 18 February to 20 May. Y-axis: The proportion of topics in the whole documents.

## 5. Discussion and Conclusions

This study explored how the topic COVID-19 was presented in the mainstream media in South Korea and how the findings can be a model for the use of statistics as an educational implementation. Here are the implications for the findings. First, topic modeling analysis was performed by extracting news articles including "Corona" and "Statistics". Throughout this process, students can collect atypical data of topics of interest becoming a recent issue using engineering tools. Setting a subject in a statistics project class can be freely performed by students or compared and discussed by a group based on the given subject. Yang [56] argues that the key in statistics project class is setting up a common goal for students to reach, then they can think and discuss the conclusions and implications. Therefore, two search keywords, "Corona" and "Statistics", were used in this study and can be utilized effectively to grasp current crisis situations that have not been experienced. More especially, *BigKinds* enables students to easily collect news articles as text data from 54 news daily outlets in South Korea with a single click. From 2018, "software class" has been a

required course in South Korea to improve computational thinking skills for students from elementary to high school curriculum [6]. As programming plays a key role in understanding software usability, it is possible to administer convergence education using Python or the R program, which students can easily access [57].

Second, four topics regarding statistics related to COVID-19, such as macroeconomy (Topic 1), domestic (Topic 2), international outbreak (Topic 3), and real estate and stocks (Topic 4), were revealed according to topic modeling analysis with semantic network analysis and clustering with structural equivalence measure. Based on this, statistics can be merged into the topics elicited from analyzing the topic modeling in statistics project class by reflecting what Douglas [58] suggested: authenticity, applied learning, and active exploration. For example, students, as democratic citizens, can improve critical thinking and reasonable decision-making skills through converging statistics and social science subjects in the process of solving the problems shared by society [59]. More specifically, it is possible to assimilate topics such as statistics, real estate, and stocks. Unlike the U.S., Canada, and England, finance education in South Korea is not required, and simple contents such as asset management or daily life as a consumer is repetitively taught in elementary, middle, and high school through sociology, technology, and the domestic science classes. Experts emphasize that systematic finance education in early ages can build up resistance against finance crisis. It is reported that even though South Korean students' mathematics scores in the Programme for International Student Assessment (PISA) are generally high, they have a shallow understanding of finance. It is said that this is because mathematics education is not fully fused with finance [60]. Therefore, through the integration of statistics and the current finance curriculum, students can be better equipped with the financial and statistical knowledge necessary for the process of decision making. In other words, students can have an opportunity to reconsider statistical value and utility in the process of problem solving and decision making in financial situations, using statistical functions and conceptions in financial problem solving, and using financial context in learning statistics.

In addition, by integrating topics related to statistics and outbreaks of disease such as COVID-19, we can expand the area of practicing statistics education to social justice. Applying a subject related to social justice such as disease, as a social issue, to statistics education helps students have a broader view of statistics and strengthens their problem-solving skills, which naturally motivates students to study statistics, provides an answer to why to study statistics, and improves critical thinking and problem-solving skills [61–64]. Even though there is not an explicit statement of social justice in the 2015 revised mathematics curriculum in South Korea, mathematical activities help students exhibit sincere, fair, and responsible behavior, and have a brave mind to overcome difficulties, an attitude of taking others into consideration, to respect and cooperate with others, and express an idea based on logical ground and practice logical decision-making skills [6]. This means that for high-quality statistics education, students should be provided with an equal chance of learning opportunity and introduced to various subjects and social issues. Therefore, statistics education for social justice that integrates statistics and disease outbreaks such as COVID-19 helps students improve their desirable personality and positive attitude throughout the mathematical process of exploring various social phenomena [65–67], which finally helps to change the negative attitude that Korean students have about mathematics [68,69].

Third, rising topics regarding the statistics related to COVID-19 were both macroeconomy (Topic 1) and real estate and stocks (Topic 4), and falling topics were both domestic (Topic 2) and international outbreak (Topic 3) as time passes based on simple linear regression analysis. Simple linear regression analysis is used to predict the future and make an objective decision based on information available from the past and the present. In South Korea, education about correlation analysis was continued until the 7th curriculum but discontinued in 2007 as a way of reducing the private education related to mathematics learning load. After that, the 2009 revised curriculum added correlation as a sub-topic of real-life topics related to statistics, which corresponds to a global trend of interpreting and critically evaluating probabilistic phenomena and statistical information under various

contexts and being able to communicate and discuss with others based on statistical information. However, unlike China, England, Singapore, Australia, and other advanced countries that include simple linear regression analysis as one of the curriculum contents, it has never been dealt with in the South Korean mathematics curriculum. Therefore, to make a connection with the teaching of correlation analysis, the subsequent revised high school mathematics curriculum needs to have regression analysis [70]. Even though there is not a direct suggestion, the 2015 revised curriculum allows teachers to use engineering tools that can analyze data and reflect academic rigor to achieve a curriculum goal without posing a learning burden [58]. Hence, in statistics project class, this can be realized by using engineering tools. For example, to predict the number of deaths by time resulting from COVID-19 by using excel or Tongrami (software developed by National Statistical Office for statistical learning) as a simple linear regression analysis tool, a sequential connectivity for correlation coefficients and regression related to the teaching correlation concept can be strengthened.

In sum, keyword analysis from term frequency revealed that the most frequent words were "confirmed case", "death toll", "mask", "job", "Italy", and "workers". The LDA topic model and semantic network analysis depicted the probability of the top 20 words belonging to each topic, where it showed how the trend of the topic changed over time. The most interesting findings were that general election issue was related to the topic of real estate and stocks. The real estate policy can vary depending on which party wins the election, so that a key word, "real estate and stock", was closely related to the national election that was administered for the first time in the world even under the pandemic. Without statistical analysis, there should have been many questions that are hard to answer regarding various social issues under COVID-19. As was shown throughout the process of this research, a statistical analysis provides implications for statistics education, such as a statistics project class which can be led by students' activities through data collection, analysis, and interpretation. With the correct implementation of statistics in the classroom, students are given a tool to make the best decision in understanding contemporary issues and preparing for an unpredictable future. The findings imply how the education curricula are to be created and implemented from the perspective of an educational sustainability in practice.

As for the limitations and suggested future directions of this study, the news articles used in the study were reported only in South Korea. It does not give an overall picture of what has been said around the world. In addition, the study has a limitation that it did not include other kinds of social media such as Twitter, Facebook, and blogs. Therefore, it is recommended to observe and analyze those kinds of data using various domestic media and international data with the same search keywords, "Corona" and "Statistics". However, given that South Korea is assumed to be one of the few countries that is successfully dealing with the unpredictable situation caused by the virus, we think that this study can show meaningful data of what people in South Korea have in their mind. Additionally, the study results are from analyzing everyday press report data from 18 February to 20 May 2020. Therefore, the results could show a narrow spectrum of contemporary issues. This limitation requires us to undertake broad data collection ranges and periods to have a better idea of the trends of the topic.

**Author Contributions:** Conceptualization, S.K. (Seokmin Kang) and S.K. (Sungyeun Kim); methodology, S.K. (Seokmin Kang) and S.K. (Sungyeun Kim); software, S.K. (Sungyeun Kim); validation, S.K. (Seokmin Kang) and S.K. (Sungyeun Kim); formal analysis, S.K. (Seokmin Kang); investigation, S.K. (Sungyeun Kim); resources, S.K. (Seokmin Kang); data curation, S.K. (Seokmin Kang); writing—original draft preparation, S.K. (Seokmin Kang) and S.K. (Sungyeun Kim); writing—review and editing, S.K. (Seokmin Kang) and S.K. (Sungyeun Kim); supervision, S.K. (Sungyeun Kim); project administration, S.K. (Seokmin Kang). and S.K. (Sungyeun Kim); funding acquisition, S.K. (Sungyeun Kim). All authors have read and agreed to the published version of the manuscript.

**Funding:** This work was supported by the Incheon National University Research Concentration Professors Grant in 2020.

**Data Availability Statement:** The data presented in this study are available on request from the corresponding author. The data are not publicly available due to a restriction by *BigKinds*.

**Conflicts of Interest:** The authors declare no conflict of interest. The funder had no role in the design of the study; in the collection, analyses, or interpretation of data; in the writing of the manuscript, or in the decision to publish the results.

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
