# Peer review of "Lessons Learned from Topic Modeling Analysis of COVID-19 News to Enrich Statistics Education in Korea"

_sustainability, doi:10.3390/su14063240_

Round 1
Reviewer 1 Report
In order to improve the quality of your article please include the following changes/adjustments/improvements:
- There is some confusion between the idea expressed in the title (Topic Modeling of COVID-19 News and Its Implications on Korean Statistics Education) and one of the specific objectives mentioned in the article (“the study intends to determine implications of South Korean statistics education based on statistics project class by performing topic modeling on data including “Corona“ and “Statistics” as atypical text material”). It is not clear whether the article studies the implications of Statistics education or the implications on Statistics education. Clarification would be recommended.
- We recommend rephrasing the title, as the analysis based on statistical methods does not show exactly the implications of Topic Modeling of COVID-19 News on Statistics Education, but perhaps only the need and recommendation to include various methods of statistical analysis in the study curriculum. There are a multitude of other phenomena in the economy, society or environment that can be modeled with statistical methods, and the example in this article is only one of many others.
- The example presented in the article is just one of the examples in which statistics can be used in everyday life, but there are many other such examples everywhere. This example alone seems insufficient to justify changing the curriculum of mathematics/statistics studied in the Korean education system.
- It is not clear the motivation for choosing the period to which the data collected refers (Feb. 17 - May 20), respectively the reasons why the authors considered this to be "the most critical and suitable period for data collection", as there were and later dramatic periods in the evolution of the pandemic in this country.
- Some sources indicate January 20, 2020 as the date of the first confirmed case of coronavirus in South Korea, while on February 17 (the date indicated by the authors as the date when the first coronavirus case was confirmed in Korea) there were already recorded 30 cases. (https://www.worldometers.info/coronavirus/country/south-korea/; https://www.ncbi.nlm.nih.gov/pmc/articles/PMC7104685/). It is recommended to check the accuracy of the data.
- In section 3.1 (Date) the variables used in the analysis should be specified more clearly.
- There is a mismatch between Topic 1 as defined in Figures 1 and 4 (In figure 1 – Topic 1 is Microeconomy, in Figure 4 – Topic 1 is Macroeconomy).
- The estimated trends of the four topics correspond to the analyzed period (Feb. 17 - May 20). The authors can complete the analysis by showing to what extent forecasts can be made for the next period, starting from the identified trends.
- To confirm the validity of the regression model, Simple Linear Regression analysis for time series must be completed (eg with serial correlation testing, residual hypothesis testing, etc.)
Author Response
Dear anonymous reviewers,
Thank you for your valuable and constructive comments.
Our responses are inserted in the table below.
We have edited and revised the draft according to your comments.
Thank you again for your time and insightful review.
Response to Reviewer 1 Comments
Point 1: There is some confusion between the idea expressed in the title (Topic Modeling of COVID-19 News and Its Implications on Korean Statistics Education) and one of the specific objectives mentioned in the article (“the study intends to determine implications of South Korean statistics education based on statistics project class by performing topic modeling on data including “Corona“ and “Statistics” as atypical text material”). It is not clear whether the article studies the implications of Statistics education or the implications on Statistics education. Clarification would be recommended.
Response 1: Thank you for your helpful comment. The second instance you mention was revised to “on,” as our research goal is to provide suggestions to Korean statistics education based on topic modeling analysis. The rest of the phrases were expressed correctly based on our research goal.
Point 2: We recommend rephrasing the title, as the analysis based on statistical methods does not show exactly the implications of Topic Modeling of COVID-19 News on Statistics Education, but perhaps only the need and recommendation to include various methods of statistical analysis in the study curriculum. There are a multitude of other phenomena in the economy, society or environment that can be modeled with statistical methods, and the example in this article is only one of many others.
Response 2: Thank you for the valuable comment. Based on your suggestion, we changed the title to “Lessons Learned from Topic Modeling Analysis of COVID-19 News to Enrich Statistics Education in Korea” so that the title focuses more on statistics education in Korea.
Point 3: The example presented in the article is just one of the examples in which statistics can be used in everyday life, but there are many other such examples everywhere. This example alone seems insufficient to justify changing the curriculum of mathematics/statistics studied in the Korean education system.
Response 3: Thank you for pointing out this weak justification. We have mentioned the limitation of implementing the statistics project method alone. At the same time, we have emphasized its great impact on mathematics education (pp.2–3).
Point 4: It is not clear the motivation for choosing the period to which the data collected refers (Feb. 17 - May 20), respectively the reasons why the authors considered this to be "the most critical and suitable period for data collection", as there were and later dramatic periods in the evolution of the pandemic in this country. Some sources indicate January 20, 2020 as the date of the first confirmed case of coronavirus in South Korea, while on February 17 (the date indicated by the authors as the date when the first coronavirus case was confirmed in Korea) there were already recorded 30 cases. (https://www.worldometers.info/coronavirus/country/south-korea/; https://www.ncbi.nlm.nih.gov/pmc/articles/PMC7104685/). It is recommended to check the accuracy of the data..
Response 4: We apologize for the confusion caused by a mistake in writing. It should include "Daegu" in the sentence. and the date needs to be changed into Korea standard time, 18 February. In addition, May 20 was selected to cover the first three months of the first massive infection wave. (with 93–94 days equaling three months). An infection wave is typically distinguished by the rapid increase of infections and following the flattened infection curve. The first wave was occurred during the three-month period in South Korea. Thus, we have revised the sentence as follows.
“The first three months of the surge of infection were the most critical in the sense that the nation experienced global pandemic. We believe it is sufficient to collect the data for three months to understand how the covid-19 has impacted the country.”
“18 February was the first day that the confirmed case of COVID-19 was identified in Daegu, Korea, where the massive virus contagion was initiated, leading to the three- month-long first massive infection wave”.
Point 5: In section 3.1 (Date) the variables used in the analysis should be specified more clearly.
Response 5: Thank you for the comment. In fact, we have explained the "date" variable in Section 3.2. We have added the following to the last sentence of Section 3.2 as follows:
"The unit of the regressor variable is a single day"
Point 6: There is a mismatch between Topic 1 as defined in Figures 1 and 4 (In figure 1 – Topic 1 is Microeconomy, in Figure 4 – Topic 1 is Macroeconomy).
Response 6: Thank you for the comment; we apologize for this mistake. We have revised topic 1 in Figure 1 to "Macroeconomy"
Point 7: The estimated trends of the four topics correspond to the analyzed period (Feb. 17 - May 20).
Response 7: Yes. The estimation is based on the period
Point 8: The authors can complete the analysis by showing to what extent forecasts can be made for the next period, starting from the identified trends. To confirm the validity of the regression model, Simple Linear Regression analysis for time series must be completed (eg with serial correlation testing, residual hypothesis testing, etc.)
Response 8: Thank you for the comment. While we understand the forecasting part of the review, we also think that analyzing the first wave would be interesting to see where the points of interest are located. In addition, unlike the time series analysis in the quantitative analysis, the regression analysis we performed addresses the frequency proportion. Thus, it is not a continuous trend with autoregressive features as time-series data typically. Due to the characteristics of the data set such as that the number of documents for each day is different, we believe that the regular time-series analysis would not fit for forecasting.
Reviewer 2 Report
-There is a need for a better rationale to link Covid-19, statistics education, and 4th industrial revolution.
-The flow in the introduction section can be improved. I mean, the section following the findings are really interesting. However, the earlier section includes many unnecessary background information. I suggest authors to refine this section to provide a smoother reading experience.
-For benchmarking purposes, authors can report the software used to conduct the analysis and visualize the figures.
-I suggest improving suggestions and implications of the study.
-Overall, the article is interesting and can potentially contribute to the related literature. I believe that the above suggestions will improve the overall quality of the paper.
Author Response
Dear anonymous reviewers,
Thank you for your valuable and constructive comments.
Our responses are inserted in the table below.
We have edited and revised the draft according to your comments.
Thank you again for your time and insightful review.
Response to Reviewer 2 Comments
Point 1: There is a need for a better rationale to link Covid-19, statistics education, and the 4th industrial revolution.
Response 1: Thank you for your comment. The concept of fourth industrial revolution was used as a task that our young generation must manage. Moreover, thinking skills embedded in statistics education are essential to prepare for an unpredictable future. In this study, we used topic modeling as a testing bed that manifests how statistics project methods can be used by adopting a topic during the COVID-19 era. This has been clarified on p.3.
Point 2: The flow in the introduction section can be improved. I mean, the section following the findings are really interesting. However, the earlier section includes many unnecessary background information. I suggest authors to refine this section to provide a smoother reading experience.
Response 2: Thank you for your valuable comment. Following your suggestion, we have deleted the less relevant sentences mainly p.1 through p.3 and refined the introduction section.
Point 3: For benchmarking purposes, authors can report the software used to conduct the analysis and visualize the figures.
Response 3: Thank you for the comment. We have added the following in Section 3.2. "We used the R software to conduct the analysis and the R software and UCINET to visualize our result".
Point 4: I suggest improving suggestions and implications of the study.
Response 4: Thank you for the comment. The suggestions and implications of the study were revised, particularly with a focus on deleting less relevant statements to deliver clear implications.
Point 5: Overall, the article is interesting and can potentially contribute to the related literature. I believe that the above suggestions will improve the overall quality of the paper.
Response 5: Thank you for your valuable comments. We have revised the draft by providing a better rationale to link the topics, clarifying the software that we used, and enriching suggestions according to your suggestions.
Round 2
Reviewer 1 Report
No comments.